# Genome of the world's smallest flowering plant, *Wolffia australiana*, helps explain its specialized physiology and unique morphology

Halim Park[1,8], Jin Hwa Park[1,8], Yejin Lee[1], Dong U Woo[1], Ho Hwi Jeon[1], Yeon Woo Sung[1], Sangrea Shim[2,3], Sang Hee Kim [1,4], Kyun Oh Lee[1,4], Jae-Yean Kim[1,4], Chang-Kug Kim [5], Debashish Bhattacharya[6], Hwan Su Yoon [7✉] & Yang Jae Kang[1,4✉]

Watermeal, *Wolffia australiana*, is the smallest known flowering monocot and is rich in protein. Despite its great potential as a biotech crop, basic research on *Wolffia* is in its infancy. Here, we generated the reference genome of a species of watermeal, *W. australiana*, and identified the genome-wide features that may contribute to its atypical anatomy and physiology, including the absence of roots, adaxial stomata development, and anaerobic life as a turion. In addition, we found evidence of extensive genome rearrangements that may underpin the specialized aquatic lifestyle of watermeal. Analysis of the gene inventory of this intriguing species helps explain the distinct characteristics of *W. australiana* and its unique evolutionary trajectory.

[1] Division of Bio & Medical Bigdata Department (BK4 Program), Gyeongsang National University, Jinju, Republic of Korea. [2] Department of Chemistry, Seoul National University, Seoul, Korea. [3] Plant Genomics and Breeding Institute, Seoul National University, Seoul, Korea. [4] Division of Life Science Department, Gyeongsang National University, Jinju, Republic of Korea. [5] Genomics Division, National Academy of Agricultural Science (NAAS) Rural Development Administration, Jeonju, Korea. [6] Department of Biochemistry and Microbiology, Rutgers University, New Brunswick, NJ, USA. [7] Department of Biological Sciences, Sungkyunkwan University, Suwon, Korea. [8]These authors contributed equally: Halim Park, Jin Hwa Park. ✉email: hsyoon2011@skku.edu; kangyangjae@gnu.ac.kr

The watermeal *Wolffia australiana* is the smallest monocot plant known to date and has a high total protein content that varies from 20% to 30% of the freeze-dry weight[1]. The latter makes *Wolffia* an attractive protein source for human or animal consumption and a factory for generating engineered peptides[2,3]. Under the vegetative reproduction mode, a floating granule frond divides once every 1–2 days. This is the fastest known plant growth rate and allows *Wolffia* to cover a pond within a few weeks[4]. This trait suggests that *Wolffia* can be developed as a biotech platform, comparable to yeast[5].

*Wolffia* is a member of the Lemnaceae in the Araceae, however, its morphology makes it distinct from sister species. Under vegetative growth, *Wolffia* occurs as a granule frond that lacks the typical plant organs such as a root, stem, or leaf. "Budding off" is the main mode of *Wolffia* reproduction, although flowers have been rarely reported[6] suggesting that *Wolffia* retains the genes needed for floral development. To compensate for the broadscale absence of sexual reproduction, *Wolffia* sinks to the bottom of lakes (i.e., turion) to over-winter. Previous studies have shown that the turion phase contains high amounts of starch and sugar that act as storage resources, whereas the summer phase is highly vacuolated and contains less starch[7]. In addition, the turion phase has thicker cell walls[7] that provide protection from frost[8]. These observations suggest that aquatic plants such as *Wolffia* may undergo atypical genome evolution that provides clues to adaptation to an aquatic lifestyle when compared to terrestrial plants.

Here, we present the draft nuclear genome of *W. australiana* generated using long-read PacBio sequence data and the 10x Genomics platform. Among *Wolffia* species, including *W. brasiliensis* (~800 Mb), *W. globosa* (~1.3 Gb), and *W. arrhiza* (~1.9 Gb)[9], *W. australiana* has the smallest genome size (~400 Mb)[9,10]. Using transcriptome data to validate predicted gene models, we created a robust gene inventory for *W. australiana*. These new genome data were compared to published genomes from two duckweed sister species within the Lemnaceae, *Spirodela polyrhiza*[11] and *Lemna minor*[12] (the *L. minor* genome was not included in downstream analyses due to its low quality). During the summer, *Wolffia* species dominate ponds with duckweeds in the genera *Spirodela* and *Lemna*. These coexisting plants share similar reproductive and turion-forming features, but only *Spirodela* or *Lemna* produce adventitious roots. Therefore, we postulate that comparative genomics among these sister species could provide clues to the unique developmental features of *W. australiana*. Differential gene expression analysis of the floating and submerged phases suggests the occurrence of hypoxia resistance associated with endoplasmic reticulum (ER) transport. More broadly, we studied the *W. australiana* genome to understand the dramatic morphological evolution and environmental adaptations that have occurred in this species that lacks typical plant-like tissues.

## Results and discussion

**Assembly of the *W. australiana* genome.** *W. australiana* is usually found together with other floating duckweed species such as *Spirodela* and *Lemna* (Fig. 1a). The genome size of *W. australiana* is ~432 Mb based on flow-cytometric genome size estimation[9,10]. Consistent with this prediction, we confirmed the genome size to be 440.27 Mb using *K*-mer analysis (Supplementary Fig. 1). For the genome assembly, we used a total of 75 Gb of PacBio data (>150× coverage; 5 million reads) with an average read length of 13,417 bp (Supplementary Data 1). The PacBio-only assembly contained 1757 high-quality contigs, totaling ~456 Mb. This exceeds the expected genome size of 440.27 Mb, therefore we used self-Blast to inspect the assembly for potential artifactual duplications. This analysis showed that 61 contigs were

duplicate entries in the assembled genome with approximately the same size (>98%) and near-perfect alignment (>98%), and a total size of ~5 Mb (Supplementary Fig. 2). We removed these duplicated contigs from the final contig set and the N50 was found to be 734,533 bases (Supplementary Data 2).

Furthermore, with ~61 Gb of mate-paired Illumina sequences, we scaffolded the PacBio contigs using SSPACE[13]. The N50 of scaffolds was increased to 888 Kb and the number of scaffolds was reduced to 1611. In addition, with ~20 Gb of Illumina data from a 10X Chromium Genome v2 library, we linked the scaffolds into super-scaffolds using ARCS[14]. The resulting 1508 super-scaffolds had an N50 = 1,169,370 bases. Notably, the value of N10 was significantly increased when compared to other scaffold size classes, suggesting that we require additional sequencing using Illumina mate-pair data and 10X Chromium Genome library to link the smaller scaffolds to ultimately generate the 20 chromosome-containing scaffolds.

The quality and coverage of the current genome assembly were assessed (Supplementary Fig. 3)[15] showing that >93% of the Viridiplantae dataset of BUSCO was detected as complete proteins. We examined the correct mapping of the paired-end reads to the assembled genome. Out of 276 million Illumina, paired-end sequencing reads, ~270 million mapped with the expected insert size.

Comparison with the recently published genomes of *W. australiana* strains 7733 and 8730 (N50 value, Wa7733: 695Kb, Wa8730: 103Kb)[16] showed that our assembly of *W. australiana* 8730 has a higher N50 value (N50, 1.17 Mb). Although the use of mate-pair and 10X platform analyses aided in longer scaffolding of the PacBio and Illumina data, additional sequence data are needed to increase completeness of the genome. This is exemplified by the recently published chromosomal-level assembly of *Spirodela* that used Oxford Nanopore and Hi-C methods[17–19]. We also compared the genome assemblies[16] to determine if they align well and if extra scaffolds or contigs exist (Supplementary Fig. 2b). Even though there are some size differences between homologous scaffolds, both genomes could be aligned well. There were 212 scaffolds in our assembly that were not paired with any Wa7733 scaffolds totaling ~13 Mbp (Supplementary Fig. 2c). The scaffold size distribution revealed a modal peak of 60 kbp and the scaffolds contained an additional 325 genes that were classified by KEGG as enzyme, transporter, chromosome and associated proteins, among others (Supplementary Fig. 2d).

### Genome annotation

*Repetitive sequence profiling.* Before gene prediction using the scaffold sequences, we used Repeatmodeler to build the repeat library specific to *W. australiana*[20]. This analysis revealed that 5.03% and 33.92%, respectively, of the *W. australiana* genome, was comprised of DNA transposons and retrotransposons (Supplementary Data 3). To elucidate the composition of repetitive sequences, we compared the data with the high-quality *S. polyrhiza* and *Oryza sativa* genomes (Supplementary Data 3). The ratio of repeated regions to the genome size of *W. australiana* (61.6%) was higher than in *S. polyrhiza* (18.5%), suggesting that the genome size difference between these species was likely to be explained by repeat content change[21]. In contrast, the percentage of retrotransposons in the total repeat content in the genome was similar between *W. austaliana* (51.3%), *S. polyrhiza* (51.6%), and *O. sativa* (56.0%) (Supplementary Data 3). However, among retrotransposons, the L1/CIN4 long interspersed nuclear elements (LINE) class showed increased representation in the total repeat content of *W. australiana* (4.1%) when compared to *S. polyrhiza* (1.9%) and *O. sativa* (2.8%) (Supplementary Fig. 3 and Supplementary Data 3). L1/CIN4 is a monocot-specific

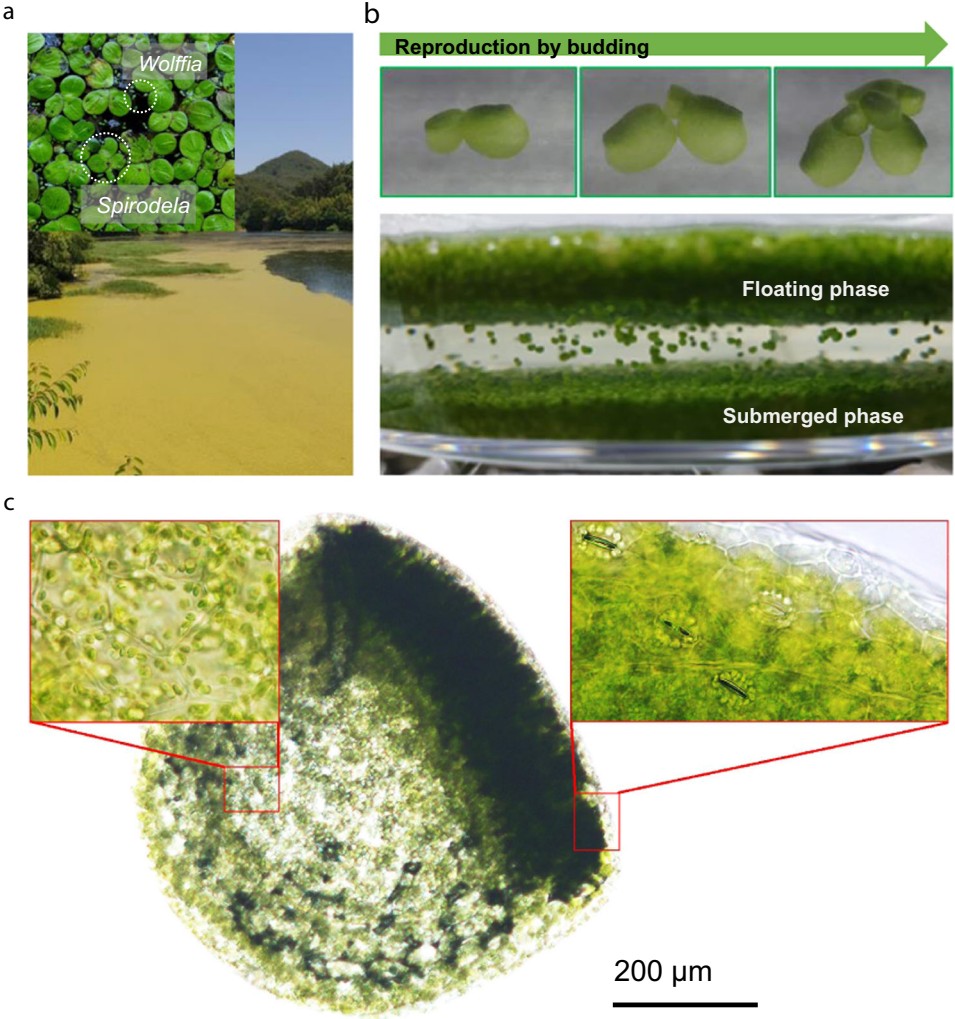

**Fig. 1 The distinct phenotype of Wolffia and its taxonomic position. a** Duckweed blooming in a lake in Jinju, Geumsan-myeon in South Korea. *Spirodela* and *Wolffia* exist together in this lake. **b** *Wolffia* growth and reproduction. **c** Micrograph of *W. australiana* showing the presence of stomata on the adaxial side and the absence of stomata on the abaxial side.

retrotransposon that may have acquired additional functionality after the divergence of dicots and monocots[22]. Moreover, L2/Cr1/Rex and R1/LOA/Jockey are present only in *W. australiana*: i.e., absent in *S. polyrhiza* and *O. sativa*. In the case of DNA transposons, the ratios to the total repeat content of the three species ranged from 5% to 37% (Supplementary Data 3). Notably, hobo-Activator showed a higher content in *W. australiana* repeats than in both sister species (Supplementary Fig. 3). From the de novo assembly of *W. australiana* genome, we calculated the composition of each transposable element class. However, it is still unclear whether transposable elements participated in the drastic phenotypic changes in *Wolffia* species. To address this issue, it is necessary to obtain a *Wolffia*-specific understanding of transposable element evolution using additional genomes from this genus.

*Gene prediction and assessment.* After masking the annotated repetitive regions in the genome, we predicted genes in the assembled genome using ab initio and homology-based methods with RNAseq data (Supplementary Data 1). The total number of predicted genes is 22,293 and the total number of proteins is 32,457, including splice variants. The average length of proteins was 396 aa, with a standard deviation of 354 aa. We assessed the predicted gene and transcript set using BUSCO and found a high coverage of known conserved genes with >94.7% and >76.3% for Viridiplantae and Liliopsida datasets as complete, respectively (Supplementary Fig. 3). The results of the BUSCO test on the Liliopsida dataset are comparable to the Wa7733 assembly which showed 69% completeness (Supplementary Data 2). For the functional annotation and phylogenetic classification of the predicted genes, we used the Eggnog database and a total of 26,205 (80.7%) out of 32,457 predicted proteins were successfully annotated[23].

**Phylogenetic relationship and divergence time estimation**. To infer phylogenetic relationships and divergence time, we reconstructed a phylogenetic tree using Bayesian inference in BEAST[24]. Six conserved nuclear genes were used in the analysis that includes eight dicots, 12 monocots including three aquatic species, one early-diverged angiosperm (Nymphaeales), one lycophyte, and one moss species (Fig. 2a). In this tree, three aquatic species (i.e., *W. australiana*, *S. polyrhiza*, *Zostera marina*) were monophyletic (the Alismatales) that diverged early from the major monocot clade. These aquatic species, however, were not clustered together with another aquatic water lily species *Nymphaea colorata*. This is explained by the early divergence of water

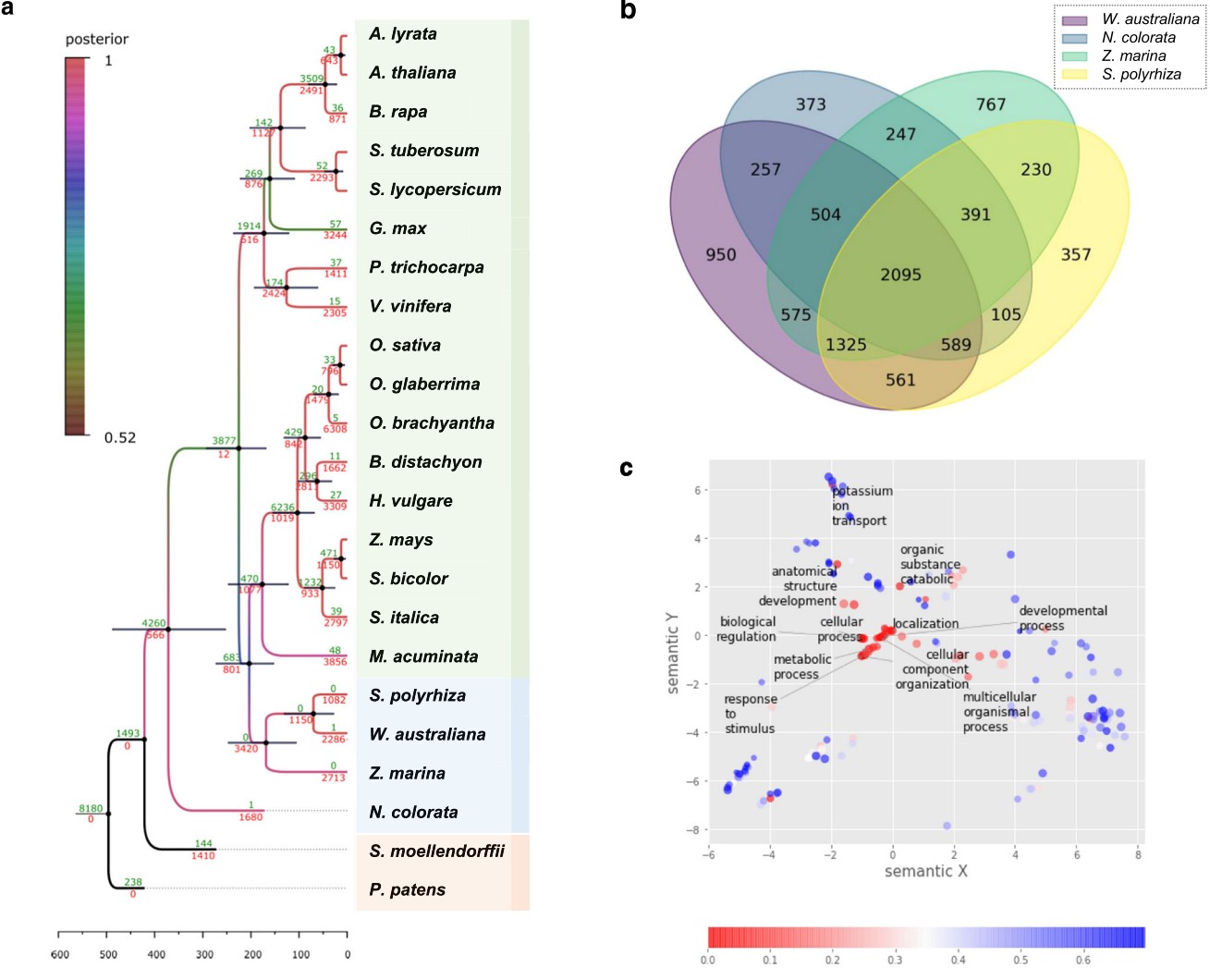

**Fig. 2 Gene family evolution in the aquatic and land plants. a** Bayesian tree based on six highly conserved proteins. The number of gene cluster gains and losses are shown in green and red numbers, respectively, at each node. The bars at nodes represent the 95% highest posterior density interval. **b** Venn diagram among the gene family losses of the aquatic plants. **c** GO enrichment analysis of *W. australiana*-specific gene family losses.

lilies among flowering plants, that is followed by the split of Amborellales[25].

For the divergence time estimation, we set the root divergence time of mosses as 496 MYA[26]. The estimated divergence time between *S. polyrhiza* and *W. australiana* was 70.22 MYA (confidence interval = 140.44–31.2 MYA), which is consistent with a previously reported split time of 73.4 MYA between genus *Spirodela* and *Wolffia* among Alismatales[27].

**Gene family evolution in aquatic plants**. Based on the assumption that the distinctive phenotype of *Wolffia* species is explained by gene content, we analyzed gene family gains/losses using the reference Bayesian tree (Fig. 2a). We applied the Dollo-parsimony (dollop) algorithm in PHYLIP using the binary scoring matrix based on the Eggnog database[28]. After collecting the number of gene family losses at each node from the dollop analysis, we compared gene family loss among aquatic plants (Fig. 2b). Compared to the larger gains in other flowering plants, it is notable that "aquatic" plants consistently lost gene families. Among these lost genes, a total of 2095 families have been shared by *Z. marina*, *S. polyrhiza*, *W. australiana*, and *N. colorata*. This trend indicates the unique trajectory of gene family evolution among aquatic plants, even though water lily (*N. colorata*) and

the three Alismatales species have independently evolved this lifestyle.

A total of 561 gene family losses occurred in watermeal and duckweed (*W. australiana*, *S. polyrhiza*), suggesting additional gene inventory size reduction in the Lemnoideae that lack stem or leaves, with vegetative, budding propagation. In terms of *Wolffia*-specific losses, 950 gene families in this category showed enrichment of GO terms, including "developmental process", "anatomical structure development", and "biological regulation" suggesting these absent functions may help explain the unique phenotype of *W. australiana* (Fig. 2c).

**Gene family expansion of water floating plants**. To study gene family expansion in aquatic species, we built a gene cluster count matrix from Eggnog DB rather than using the presence/absence matrix (Supplementary Data 4). We used the random forest classification algorithm to identify gene expansion in floating plant species. Based on these results, 34 informative clusters were extracted and analyzed (Fig. 3a). Among these clusters, dehydrogenases, such as alcohol dehydrogenase (EC:1.1.1.1), S-(hydroxymethyl)glutathione dehydrogenase/alcohol dehydrogenase (EC:1.1.1.284 1.1.1.1), alcohol dehydrogenase class-P (EC:1.1.1.1), and 17-beta-estradiol 17-dehydrogenase/very-long-chain 3-oxoacyl-CoA reductase (EC:1.1.1.62

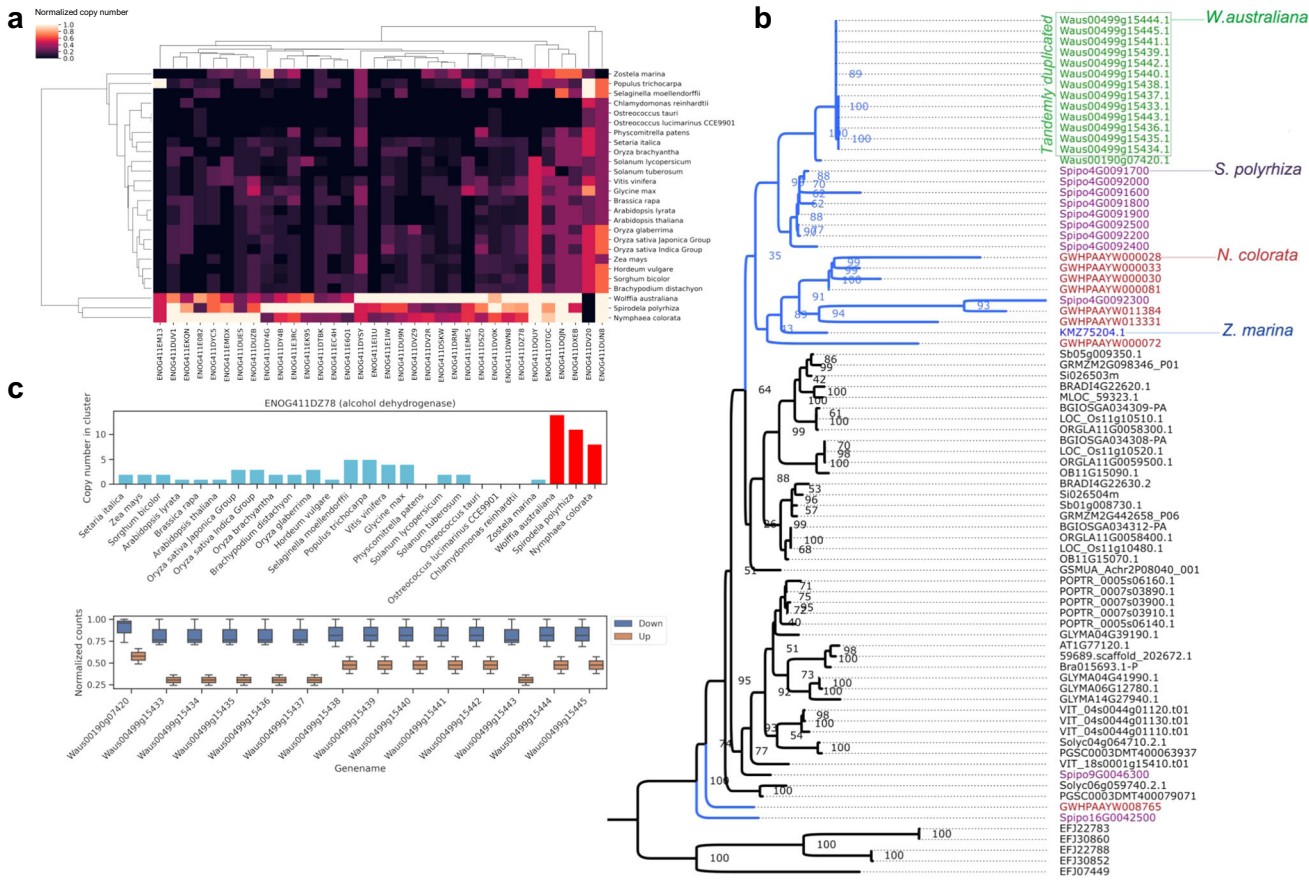

**Fig. 3 Gene family evolution in floating plants. a** Heatmap of selected Eggnog clusters that classify floating plants. **b** Gene tree of alcohol dehydrogenase (ENOG411DZ78). The blue line indicates aquatic plants with their species names being distinguished by different colors. Tandemly duplicated genes in the *W. australiana* genome are indicated by the green box. The gene IDs, which indicate the corresponding scaffold and gene order number, can be used to identify tandemly duplicated loci. **c** Copy number of alcohol dehydrogenase genes (ENOG411DZ78). The copy number in floating plants is shown with the red bars and their gene expression in the floating and submerged phases in *W. australiana* are shown in the bottom panel.

1.1.1.330) were highly expanded in the genomes of "floating" aquatic plants (*W. australiana*, *S. polyrhiza*, and *N. colorata*) that were clearly distinct from those of other flowering plants as well as a marine aquatic plant, *Z. marina*.

The alcohol dehydrogenase cluster (ENOG411DZ78) contained 14 *W. australiana* genes that were mostly tandemly duplicated (Fig. 3b). Alcohol dehydrogenases are important under root hypoxic conditions[29] and an increase in the copy number would presumably aid the survival of floating plants, particularly when submersed.

The wax and cuticle components of aquatic plants may act as protective layers against damaging UV irradiation[30]. Supporting this idea, a high percentage of phytosterol has been reported in *S. polyrhiza*[30]. Moreover, of the total lipids in *Wolffia*[31], phytosterols constitute a significant proportion, suggesting essential roles in the floating lifestyle. In line with these observations, floating plants show a larger gene family encoding estradiol 17-dehydrogenase, which is involved in cutin, suberin, and wax biosynthesis in the fatty acid elongation pathway (ko00062, Kegg Pathway)[32] (Supplementary Fig. 4).

Based on Kegg Brite classification of the expanded gene families, the remaining "enzymes" such as peroxidase, hydroxylase, dioxygenase, and kinase were highly expanded in floating plants (Supplementary Fig. 5). In addition to these enzymes, other Kegg Brite classes such as "exosome", "transporter", "ubiquitin system", "membrane trafficking", "chaperones and folding catalysts" were also expanded (Supplementary Fig. 6). Notably, the heat shock

protein 20 (HSP20) family was highly expanded in floating plants. *W. australiana* encodes 18 copies, indicating enhanced chaperone activities.

**Gene family evolution in *W. australiana*.** To understand *W. australiana*-specific gene evolution, we identified Eggnog gene clusters that are absent or over-represented in this species when compared to other flowering plants. A total of 246 and 17 clusters, respectively, were missing and over-represented only in *W. australiana* (Supplementary Data 5). Among these, the missing orthologs may contribute to the distinctive, morphologically simpler phenotype of *W. australiana* (see below).

*Root degeneration of* W. australiana. Auxin synthesis, signaling, and transport are involved in plant organ development and are well studied in the root system[33]. Auxin responsive factor (ARF) and indoleacetic acid-induced protein (IAA) occupy essential roles in lateral root development[34]. In the genome of the rootless *W. australiana*, several important components in the auxin response pathway are missing (Fig. 4a). The genes encoding IAA3, IAA12, and IAA14 are present, whereas IAA27 is absent in this genome. In the case of the ARF family, ARF7, ARF10, ARF16, and ARF19 are generally found in all plant species, however, ARF5 that positively regulates priming and lateral root initiation[34] is absent in *W. australiana* and *Z. marina*. Furthermore, ARF9, which is important for the response to auxin accumulation[35], and its homologs (ARF11 and ARF18) are absent only in *W. australiana*. Among the

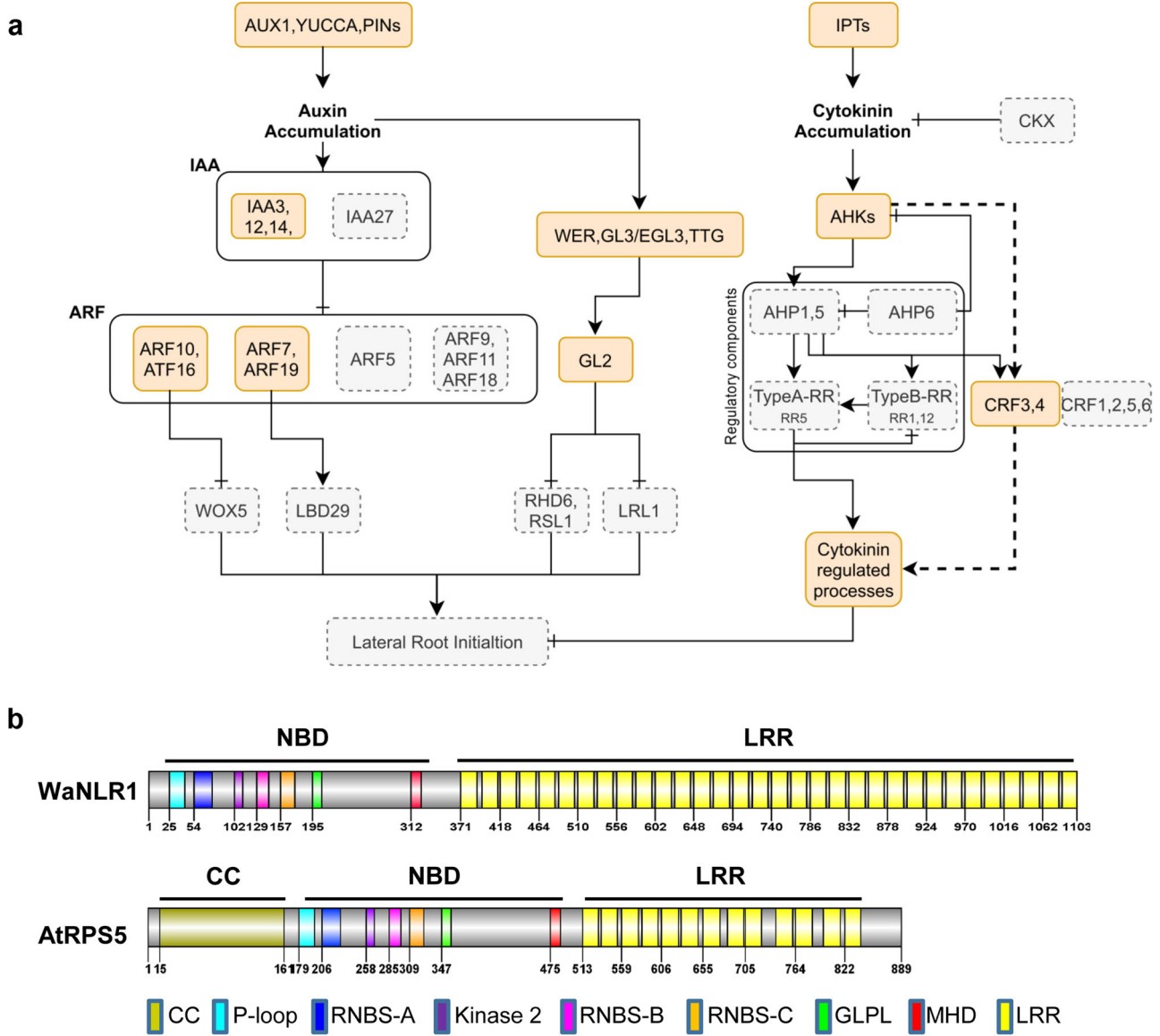

**Fig. 4 Gene family evolution of *W. australiana*. a** Gene family presence and absence in the auxin signaling pathway of *W. australiana*. The orange and gray boxes represent presence and absence of gene families, respectively. **b** The only NLR gene in the *W. australiana* genome, WaNLR1, and its structure when compared to AtRPS5.

downstream members of the ARF family, wuschel-related homeo-box 5 (WOX5) and lateral organ boundaries domain 29 (LBD29) are missing in *W. australiana* genome. WOX5 controls root stem cell niches[36] and LBD29 participates in lateral root formation, downstream of ARF7 and ARF19[37].

Another downstream enzyme in auxin accumulation is GLABRA2 (GL2), which suppresses the basic helix-loop-helix (bHLH) transcription factors, root hair defective 6 (RHD6) and Lj-RHL1-like1 (LRL1), that are required to form lateral roots. The *W. australiana* genome lacks RHD6 and LRL1. These results suggest that gene losses in the auxin signaling pathway, particularly downstream of ARF and GL2 may have driven root degeneration in *W. australiana*. Parallel to reduction in the auxin response pathway, gibberellin-insensitive dwarf 2 (GID2) that participates in the gibberellin pathway is absent in the *W. australiana* genome[38] (Supplementary Data 5). Gibberellin participates in many aspects of plant development and the absence of this pathway in *Wolffia* may also explain the absence of root and shoot growth[39].

Furthermore, the cytokinin signaling pathway in the genome of *W. australiana* appears to be disrupted (Fig. 4a). For instance, cytokinin oxidase (CKX) that degrades cytokinin is missing. As a consequence, increased cytokinin response could inhibit lateral root development[40]. In addition, many cytokinin signaling and regulatory components, such as response regulators (Type A-RR, Type B-RR) and histidine phospho-transferase proteins (AHP) are absent from the genome[41]. Based on this finding, we postulate that *W. australiana* may be unable to control cytokinin regulation, resulting in root degeneration.

*Adaxial stomata development of floating plants.* Unlike other land plants, the stomata of floating plants is located on the adaxial side of the leaf or frond[30,42]. In the case of *Wolffia*, a few stomata are located on the dorsal surface[43] (see Fig. 1c). In the submerged marine species *Z. marina*, stomata have been lost and this genome shows a highly diverged gene content, particularly, with respect to stomatal differentiation[44]. To understand stomatal

development in floating plants, we inspected the gene list associated with stomatal differentiation in *W. australiana* and in other plants using Eggnog DB and Eggnog annotation of the aquatic plants included in this study. The known stomata differentiation genes, including SPEECHLESS (SPCH), MUTE, and FAMA exist in the *W. australiana* genome. In contrast, we found EPIDERMAL PATTERNING FACTOR-like protein 9 (EPFL9, AT4G12970) that is commonly absent in aquatic plants, including *Z. marina*, *S. polyrhiza*, *N. colorata*, and *W. australiana* (Supplementary Data 6). In *Arabidopsis*, EPFL9 is a positive regulator of stomatal development[45] and the loss of EPFL9 may have affected the inactivation of abaxial stomata development in floating plants.

*Disease resistance of* W. australiana. The NLR [nucleotide-binding domain (NBD) and leucine-rich repeat (LRR)] genes in *W. australiana* are largely missing and only one NLR gene (Waus00068g18410.1, WaNLR1) was found. Although NLRs are not present in green algae (e.g., *Chlamydomonas reinhardtii*), bryophytes (*Physcomitrella patens*) possess 49 NLRs, and *A. thaliana* and *O. sativa* contain 182 and 438 NLRs, respectively. In the case of aquatic plants, they have been reported to have 44, 52, and 416 NLRs for *Z. marina*, *S. polyrhiza*, and *N. colorata*, respectively[11,25,44].

Even though the closest orthologs of WaNLR1 in *A. thaliana* are RPS2 and RPS5[46,47], WaNLR1 encodes a NB-LRR (NL) class NLR protein because it contains NB and LRR domains without a N-terminal motif (Fig. 4b). Well conserved NBD motifs, such as P-loop, RNBS-A, Kinase 2, RNBS-B, RNBS-C, GLPL, and MHD were found in the N-terminal region of WaNLR1, but the CC or Toll-interleukin-1 receptor (TIR) domains found in prototypical NLR proteins were absent. The C-terminal region of WaNLR1 comprises 32 imperfect LRRs, which is far greater than in RPS5 and RPS2 in *A. thaliana* (~13 LRRs). Meanwhile, the high number of LRRs is comparable to FLS2 and Cf-2, receptor-like proteins with extracellular LRR motifs containing 28 and 38 LRRs, respectively, and the LRRs were proposed to recognize corresponding pathogen molecules with their central or more C-terminal LRRs[48,49]. Therefore, we speculate that WaNLR1 may sense pathogen-derived molecules directly through its LRR domain.

*Gene family expansion in* W. australiana *is a response to environmental stress*. In addition to the missing orthologs in *W. australiana*, we found 17 gene families that show a significant increase in copy number (Supplementary Data 5). The two most notable over-represented Eggnog clusters are arabinosyltransferase (XEG113) and O-acyltransferase WSD1-like genes that have 27 and 26 gene copies in *W. australiana*, which is >10-fold higher than in the sister species *S. polyrhiza*. Arabinosyltransferase (XEG113) participates in the arabinosylation of cell wall components, which is important for cell elongation, root development, and cell wall defense[50,51], whereas O-acyltransferase WSD1-like genes participate in wax biosynthesis. Together with the estradiol 17-dehydrogenase gene expansion in water floating plants, cuticle wax biosynthesis would also be important for the survival of *W. australiana*, because, (i) the capacity of wax components to repel water would be important for *W. australiana* to float in the correct position with the stomata facing the air to aid respiration, (ii) wax cuticle layers can minimize UV radiation damage[30] and improve fitness even when exposed to direct sunlight on the water surface with no shade, and (iii) it is suggested that plant cuticles can play significant roles in plant-pathogenic and nonpathogenic relationships[52]. Collectively, these expanded gene families are likely to play a critical role in the adaptation to an aquatic environment.

**Genome evolution of *W. australiana*.** Genome synteny between *W. australiana*, *S. polyrhiza*, and *O. sativa* was studied to understand the dynamics of genome rearrangement vis-à-vis species divergence. We plotted the synteny of *W. australiana* and *S. polyrhiza* against *O. sativa* chromosome sequences with the mean Ks value of each synteny block (Fig. 5a). *S. polyrhiza* showed conserved synteny blocks when compared to *O. sativa* chromosomes, however, *W. australiana* showed a reduced number with respect to *O. sativa*. A total of 127 synteny blocks that are well-conserved between *S. polyrhiza* and *O. sativa* were missing and 159 synteny blocks were partially missing in *W. australiana* (Supplementary Data 7–9). Considering the close phylogenetic distance between *S. polyrhiza* and *W. australiana*, the high synteny loss in the *W. australiana* genome indicates rapid rearrangement. We hypothesized that synteny loss may contribute to the distinct phenotype of *W. australiana*, therefore we studied the gene ontology of "extra" genes in the fractionated synteny blocks of *O. sativa* (Fig. 5b). Interestingly, the enriched gene ontologies of the partially missing synteny were mostly development-related terms, consistent with the simple, granule-like morphology of *Wolffia*. The missing synteny showed GO terms were related to stress-responses.

We calculated the mean Ks value distribution of intra-genome synteny blocks in *W. australiana* and in sister species (Fig. 5C). The floating species, *N. colorata* and *S. polyrhiza* often retain the ancient whole genome duplication (WGD) peak ($Ks \cong 0.85$), which is older than the duplication peak of *O. sativa* ($Ks \cong 0.68$). Assuming the ancient WGD event is shared among the floating species, evidence of the ancient WGD event is missing in *W. australiana*. This result is consistent with rapid genome fractionation in *W. australiana* after WGD. Furthermore, a very recent WGD peak in *W. australiana* was found near $Ks \cong 0.02$. The total number of synteny blocks with a mean Ks value < 0.1 was 134 and they consisted of 989 gene pairs. These recent gene duplications may be explained by selection to functional versatility, or alternatively, they may be more ancient events that are being homogenized by concerted evolution[53].

**Adaptation in *W. australiana* to an anaerobic environment.** Using the genome assembly of *W. australiana*, we uncovered rapid genome evolution, including species-specific gene divergence, gene absence/expansion, and synteny loss. To study gene expression in the context of adaptation to an aquatic habitat, we used RNAseq to study the floating and submerged phase of *W. australiana* (Fig. 1b and Supplementary Data 10). The submerged phase is referred to as the "turion" that can survive in pond bottoms during the cold winter months. Three biological replicates were sequenced for each phase, however, the first biological replicate for the floating phase (Up1) was removed during the quality control process (Supplementary Fig. 7). A total of 698 differentially expressed genes (DEGs) were identified based on the Bonferroni adjusted *p*-values. Among these, 189 and 509 genes were upregulated in the floating and submerged phase, respectively (Fig. 6a, b and Supplementary Data 11). Based on GO enrichment analysis, the 189 upregulated genes in the floating phase were significantly enriched in stress responses related terms, whereas the 509 upregulated genes in the submerged phase were enriched in transport related terms (Supplementary Fig. 8).

From 509 significantly upregulated genes in the submerged phase, 286 could be annotated using Kegg Orthology (KO) (Supplementary Data 12). The alcohol dehydrogenase cluster (ENOG411DZ78) showed upregulation in gene expression during the submersed phase compared to the floating phase (Fig. 3c), therefore, *Wolffia* may survive under hypoxia/anoxia by

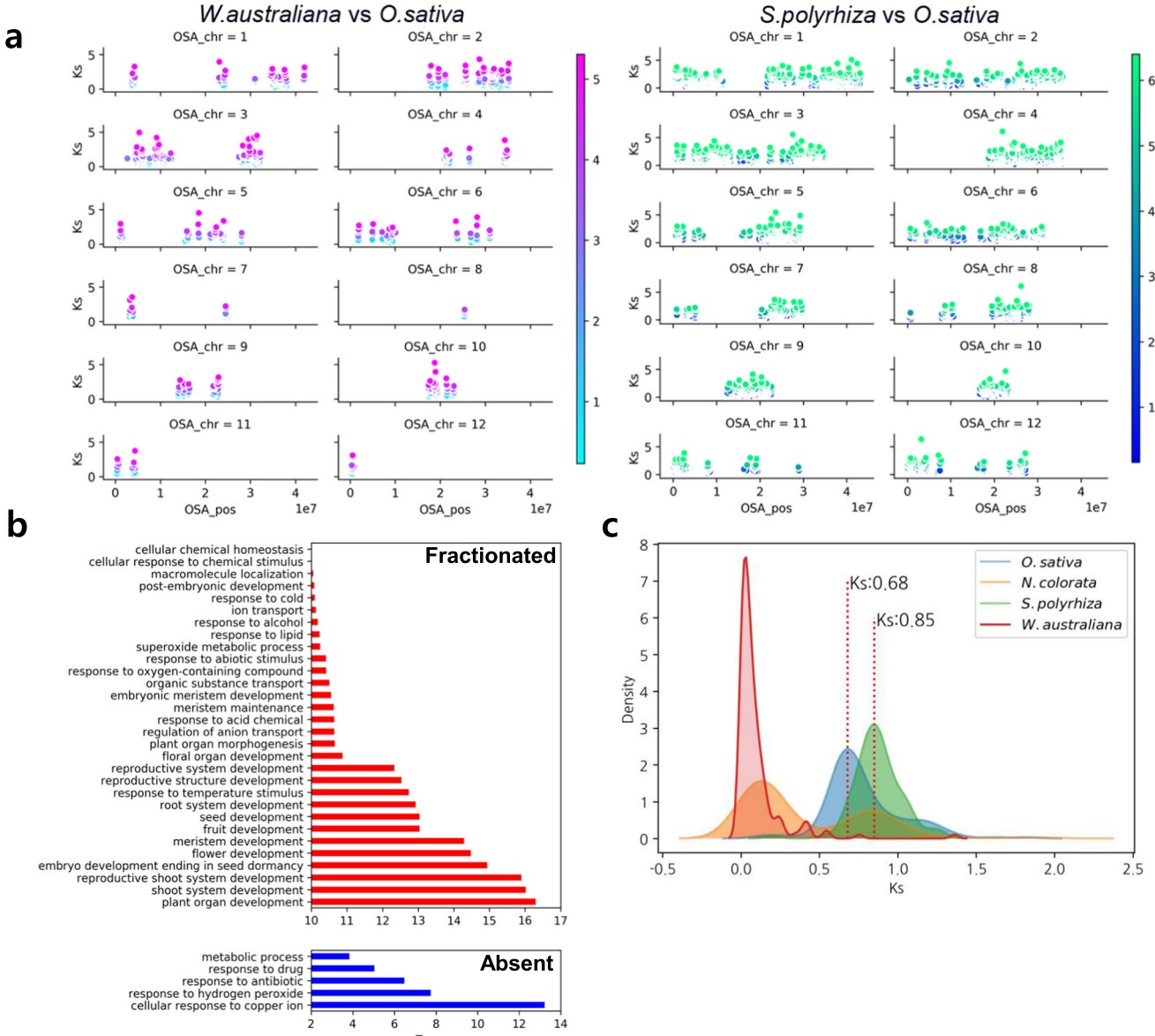

**Fig. 5 Fractionation of the synteny blocks in *W. australiana*. a** Synteny blocks were plotted on the chromosomes of *O. sativa*. The *x*-axis shows chromosomal positions and the *y*-axis shows the mean Ks values between *W. australiana* and *O. sativa* (left panel) and between *S. polyrhiza* and *O. sativa* (right panel). **b** GO enrichment analysis of the gene set from partially fractionated synteny blocks (top panel) and from missing synteny blocks (bottom panel). **c** Kernel density estimation plot of mean Ks values of intra-genome synteny blocks. The red vertical lines indicate ancient WGD peaks with the Ks values shown.

upregulating alcohol dehydrogenase[54]. Moreover, the abscisic acid signaling pathway seems to be suppressed during the submersed phase by up-regulation of the leucine-rich repeat receptor-like protein kinase cluster (ENOG411DY4B) (Fig. 6c). A gene within this cluster, IMPAIRED OOMYCETE SUSCEPTIBILITY1 (IOS1) in *A. thaliana*, is known to negatively regulate the abscisic acid signaling pathway[55].

The Kegg BRITE classification scheme showed that 133 KO were classified as "Enzyme", 33 KO were classified as "Membrane trafficking". From the Kegg Pathway mapping result, we found the genes in "Citrate cycle pathway (TCA cycle, ko00020)"; succinate dehydrogenase [EC:1.3.5.1], isocitrate dehydrogenase [EC:1.1.1.42] and phosphoenolpyruvate carboxykinase (PEPCK) [EC:4.1.1.49], which are considered to be linked to hypoxia[56,57] (Supplementary Fig. 9). Activation of PEPCK is not well documented in plants, however, multiple studies in human research report the upregulation of PEPCK under hypoxia[58,59].

Moreover, we found hypoxia upregulated 1 (HYOU1) among 10 distinct genes in "Protein processing in ER pathway (ko04141)". This gene is labeled as HSP70-17 in the *A. thaliana* annotation and has not been extensively studied. However, in animals, it is reported as being part of the cytoprotective cellular mechanism against oxygen deprivation and alleviation of ER stress[60]. Beside the hypoxia-related genes, sucrose synthase [EC:2.4.1.13], beta-glucosidase [EC:3.2.1.21], 1,3-beta-glucan synthase [EC:2.4.1.34], 1,4-alpha-glucan branching enzyme [EC:2.4.1.18] and 4-alpha-glucanotransferase [EC:2.4.1.25] in the starch and sucrose metabolism pathway (ko00500) were upregulated in the submerged phase, possibly accumulating D-glucose, sucrose, and starch. This result is consistent with the known physiology of turion formation that leads to the accumulation of numerous starch grains[61]. From these upregulated gene in the submerged phase, we gained insights into how *Wolffia* survives and overcomes hypoxia stress in pond bottoms during winter.

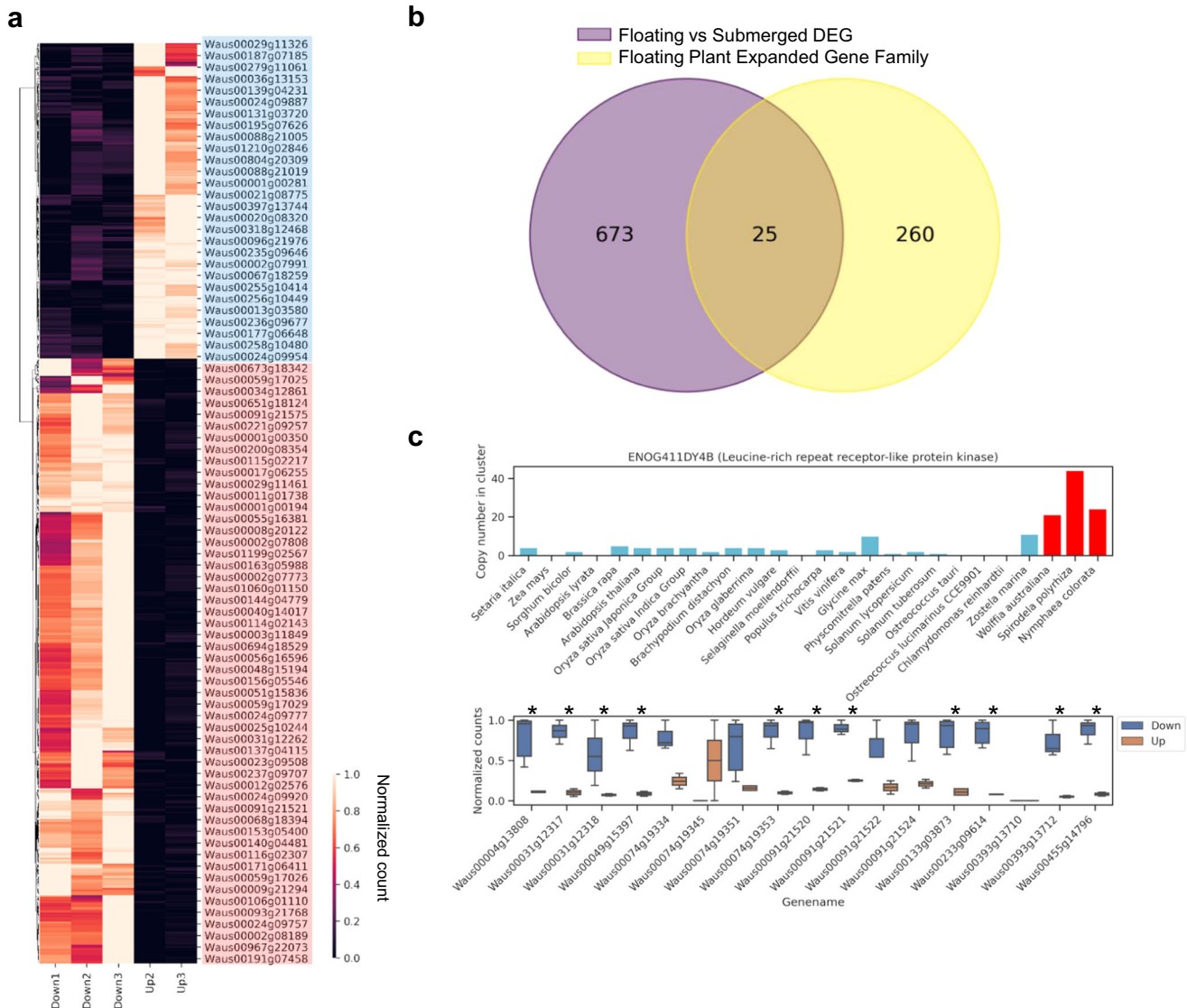

**Fig. 6 RNAseq analysis of the floating and submerged phases of *W. australiana*. a** Heatmap of DEGs that discriminate the floating (Up) and submerged (Down) phases. **b** Venn diagram showing the number of shared genes in the highly expanded gene families and the significant DEGs in *W. Australiana*. **c** Copy number of leucine-rich repeat receptor-like protein kinase cluster (ENOG411DY4B). The copy number in floating plants is shown with the red bars and their gene expression in the floating and submerged phases in *W. australiana* are shown in the bottom panel.

## Conclusions

The phenotype of *W. australiana* is unique among flowering plants. Molecular data place *W. australiana* in the Araceae, however, how this species lost the leaf and root system remains an open question. Here, we generated a high-quality nuclear genome assembly of *W. australiana*. The size of this genome is consistent with predictions using flow cytometry and *K*-mer analysis[9]. The predicted gene catalog is highly represented in the conserved BUSCO database[15].

Gene clustering, pathway mapping, synteny analysis, and other comparative approaches demonstrate rapid genome evolution in *W. australiana* with extensive gene loss and gene expansion resulting in synteny loss, when compared to other species. Specifically, the plant–pathogen interaction and hormone signaling pathways are rewired when compared to sister species. Because floating plants spend their lives in close contact with the water and often interact with micro-organisms, this may lead to the configuring of pathways to reduce sensitivity to pathogens. Surprisingly, the NBS-LRR gene family is highly reduced. However, this does not necessarily imply lowered stress resistance in *W. australiana*, rather, it may indicate

other strategies to cope with pathogens. We also inspected the basis of length growth suppression in *W. australiana*. The reduction or absence of auxin and gibberellin pathways may explain the granule form of this species. Interestingly, in the genome of the rootless eudicot *Utricularia gibba*[62], the auxin signaling pathway components including LBD29, WOX5 and RSL1, are missing.

How the apparent rapid pathway rewiring occurred in *W. australiana* remains an interesting question. We tried to address this issue using synteny analysis that compared the number of shared blocks between *S. polyrhiza* and *O. sativa* and between *W. australiana* and *O. sativa*. The partially and completely missing synteny blocks showed GO enrichment in development-related and stress-response terms, respectively, suggesting that synteny fractionation may be a driving force of rapid genome evolution[63]. Concurrently, the overrepresentation of gene families in the *W. australiana* genome with functions such as arabinosyltransferase suggest rapid, potentially adaptive duplication events.

We gained insights using RNAseq analysis that may help explain how *W. australiana* survives in the bottom of the ponds

during winter. Hypoxia-related genes and "Membrane trafficking"-related genes were upregulated in the submersed phase. HYOU1, a putative molecular chaperone that may help stabilize proteins under oxygen depletion was also upregulated in the submersed stage. In addition, the sucrose metabolism pathway was active, possibly accumulating D-glucose, sucrose, and starch, which is consistent with the known physiology of turion formation that leads to accumulation of starch grains[61]. Taken together, the genome sequence of *W. australiana* provided many insights into the unique physiology of this floating plant. With regard to the dietary and scientific value of *W. australiana*, the genome and gene inventory resulting from this study lay the foundation for future research into these important topics.

## Methods

**Plant material and culture**. For plant tissue, we acquired *W. australiana* 8730 from the Rutgers Duckweed Stock Cooperative (RDSC, http://www.ruduckweed.org/). The culture medium was composed of 0.5× SH salt, 1.5% sucrose and adjusted to pH 6. The condition of culture was in the temperature of 22 °C with the light condition of 5000 lx.

### High-throughput sequencing methods

*Sequel library construction and sequel sequencing*. Using the Covaris G-tube, we generated 20 kb fragments of genomic DNA according to the manufacturer's recommended protocol. We additionally used the AMpureXP bead purification system to eliminate small fragments. A total of 5 µg for each sample was used as input for the preparation of the sequencing library. The SMRTbell library was built using the SMRTbell® Express Template Preparation Kit (101-357-000). The SMRTbell library was sequenced using SMRT cells (Pacific Biosciences) and Sequel Sequencing Kit v3.0. Total 1 × 10-h real-time sequencing were recorded for each SMRT Cell 1M v3 using the sequel (Pacific Biosciences, PacBio) sequencing platform.

*WGS Library construction and Illumina sequencing*. We verified DNA reliability using 1% agarose gel electrophoresis and the Qubit dsDNA HS Assay Kit (Thermo Fisher Scientific). The DNA library was prepared in accordance with the recommended procedure of the Truseq Nano DNA Library kit and Nextera Mate Pair Library Prep Kit. Using the Covaris S2 system 0.2 µg of high molecular weight genomic DNA was randomly screened for sample library preparation to yield the desired size of DNA fragments. The reliability of the amplified libraries has been checked by capillary electrophoresis (Bioanalyzer, Agilent). WGS sequencing is performed using the Illumina NovaSeq 6000 and Hiseq2000 system following the 2 × 100 sequencing protocols.

*10X Chromium Genome v2 library construction and Illumina sequencing*. Before the sequencing library was constructed, we removed small DNA fragments using BluePippin size selection system and AMpureXP bead purification system to clean the gDNA for large-insert preparation. A total of 10 ng for each sample was used as input for the preparation of the library. The 10X Chromium genome v2 library was constructed using the chromium Genome Library & Gel Bead kit v2 (PN 1000017) and Genome chip kit v2 (PN 120257). 10X Genome Sequencing is performed using the Illumina NovaSeq 6000 platform following the 2×150 sequencing protocols.

**Genome assembly**. PacBio long reads are assembled into contigs using Falcon v0.3.0 after Canu v1.0 error correction[64,65]. For the assembly correction, the short reads generated by Illumina Novaseq 6000 were used. Using BWA[66], the short reads were mapped to the contigs and the variants were called by Samtools[67]. The contigs were scaffolded using SSPACE v3.0 and further super-scaffolded using ARCS with the 10X genomics data[13,14]. After each round of scaffolding, we used GapFiller v1.1 to minimize the stretches of 'N' bases[68]. These contigs were evaluated with short reads using high-quality Illumina data at high depth of coverage. We produced two batches of Illumina sequencing totaling ~88 Gb and mapped to contigs with software BWA[66] (Supplementary Data 1). The high-quality variant calls were collected and presented in Supplementary files 1 in VCF file format. The criteria for determining the high-quality were chosen from the distribution of mapping quality associated values (QUAL > 200, MQ > 60) in the VCF file (Supplementary Data 13). A total of 35,664 SNPs were obtained as high-quality homozygous variants and would be referred to as Illumina platform specific calls for *W. australiana*. We also found 126,418 heterozygous SNPs showing low heterozygosity of this accession, *W. australiana* 8730.

### Genome annotation

*Repeat masking*. For the de novo repeat element mining, we implemented Repeatmodeler[20] using RMblast engine (http://www.repeatmasker.org/RMBlast.html). For the further mining of the long terminal repeats (LTR) we allowed the LTR_Struct[69] in the repeat modeling pipeline. Together with the results from

Repeatmodeler and LTR Struct, the repeat library for *W. australiana* were successfully built and we annotated the repeat regions on the *W. australiana* genome using Repeatmasker[70]. In addition, the hmm of AP_ty1copia and AP_ty3gypsy elements was built using their alignment information from GyDB[71]. Based on the collected library of transposable elements, we masked the *Wolffia* assembly using Repeatmasker[70].

*Gene prediction*. After masking the annotated repetitive regions in the genome, Augustus-based gene prediction was used to call genes[72]. *W. australiana* specific gene prediction parameters were prepared based on the conserved gene set using BUSCO gene set, "viridiplantae_odb10"[15] and implemented gene prediction using Augustus[72] on the repeat masked genome sequences. In addition to the gene prediction, we updated the gene model based on RNAseq alignments against the genome sequences. The alignment of Illumina RNAseq data was conducted using Hisat2 program[73] and the transcript reconstruction of the alignments was performed with StringTie[74]. Comparison with the Augustus-based gene/transcript boundary prediction, we updated the reconstructed transcript assembly from RNAseq alignments as additional isoforms.

### Statistics and reproducibility

*RNAseq analysis*. Low-quality reads were trimmed using trimmomatic software[75]. Using the program Kallisto, the filtered short reads were mapped to the *W. australiana* reference coding sequences. Deseq2[76] was used to extract candidate gene set as DEG that differentiate between the floating and submerged phases using the gene expression counts. The final candidate genes were chosen based on Bonferroni modified *p*-values < 0.001.

*Phylogenetic analysis*. For the nuclear genome-based phylogenetic tree, we selected the six most highly conserved genes shared by *W. australiana*, *S. polyrhiza*, *O. sativa*, and *Z. marina* and analyzed the alignments using Bayesian tree construction in BEAST[24]. The LG model[77] was the best-fit model determined by ProtTest v3.4.2. The divergence times were relatively calculated based on the root divergence time as 496 MYA[26]. GO enrichment analysis was visualized using REVIGO program[78].

**Reporting summary**. Further information on research design is available in the Nature Research Reporting Summary linked to this article.

## Data availability

NCBI BioProject accession number for genome assembly: PRJNA611905. The raw reads of *W. australiana*, including genome and transcriptome data, can be downloaded from NCBI SRA, BioProject accession number: PRJNA734041. The genome and gene information are freely shared at https://duckweeds.plantprofile.net/.

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

## Acknowledgements

This work was supported by a grant from the Next-Generation BioGreen 21 Program (SSAC PJ013890-01, -02, -03) and the Cooperative Research Program for National Agricultural Genome Program (PJ01347303) from the Rural Development Administration, Republic of Korea. It was partially supported by the National Research Foundation of Korea (NRF-2017R1A2B3001923, 2020M3A9I4038352). D.B. was supported by a NIFA-USDA Hatch grant (NJ01180).

## Author contributions

Y.J.K., H.S.Y., C.-K.K., S.H.K., K.O.L., and J.-Y.K. designed the experiment; Y.W.S. and J.-Y.K. subcultured the strains; J.H.P., H.P., and Y.W.S. performed the RNAseq experiment; Y.L., D.U.W., H.H.J., S.S., and Y.J.K. analyzed the data; Y.J.K., H.S.Y., and D.B. wrote the manuscript. All of the authors have read and approved the final manuscript.

## Competing interests

The authors declare no competing interests.
