## [Peer Review File · Communications Biology]

Reviewers' comments:

Reviewer #1 (Remarks to the Author):

In this manuscript, authors generated the reference genome of *Wolffia australiana*, identified the genome-wide features that may contribute to its atypical anatomy and physiology, including the absence of roots, adaxial stomata development, and anaerobic life as a turion. The study provided many novel insights into the unique physiology of this floating plants with unique morphology. The work is convincing. I recommend this study if authors could respond or do some minor revisions to my following comments:

1. In the portion of "Root degeneration of *Wolffia Australiana*". The reduction or absence of auxin and gibberellin pathways were proposed to explain the absence of root and shoot growth. In table s4, "ENOG411DVJU" and "ENOG411DV05" involved in the pathway of cytokinin were missing. Additional analysis of reduction/expansion of cytokinin and other hormone pathways will make the explain more convincing. Furthermore, the other rootless species should be compared, such as *Utricularia gibba*.

2. Line 30-33 : "Watermeal, *Wolffia australiana*, is the smallest known flowering monocot that is rich in protein. Despite its great potential as a biotech crop, basic research on *Wolffia* is in its infancy and a high-quality genome assembly does not yet exist."

This is not true anymore. The genomes of two *W. australiana* strains (7733, 8730) have been published recently (doi: 10.1101/gr.266429.120). The sentence have to change a tiny bit.

3. Line 182-183: "The alcohol dehydrogenase cluster (ENOG411DZ78) contained *W. australiana* genes that were mostly tandemly-duplicated (Fig. 3B)."

The direct evidence for "tandemly-duplicated" should be provided. Then, in Fig. 3b: the genes should be marked with distinct color one by one for those involved in alcohol dehydrogenase cluster of *W. australiana*, *S. polyrhiza*, and *N. colorata*. Some genes were ignored while some were marked improper, such as SPO_Spipo4G0092300, ZmA_KMZ75204.1, SPO_Spipo9G0046300, GWHPAAYW008765...

4. Line 250-252: "Although NLRs are not present in green algae (e.g., *Chlamydomonas reinhardtii*), bryophytes (*Physcomitrella patens*) possess 49 NLRs, and *A. thaliana* and *O. sativa* contain 182 and 438 NLRs, respectively."

Add the information of other aquatic plants, such as *S. polyrhiza*, *Z. marina*, and *N. colorata*.

5. Line 273-275: "Together with the estradiol 17-dehydrogenase expansion of water floating plants, cuticle wax biosynthesis would be important for the survival of *W. australiana*."

This is an interesting finding, Please provide in-depth analysis of the relationship between the cuticle wax and the survival of *W. australiana*. i.e. protective layers against damaging UV irradiation/pathogen infection, hydrophobicity of granule frond, and keep the correct position.

6. Line 292-297 "We calculated the mean Ks value distribution of intra-genome synteny blocks in *W. australiana* and in sister species (Fig. 5C). The floating species, *N. colorata* and *S. polyrhiza* often retain the ancient whole genome duplication (WGD) peak ($K_s \cong 0.85$), which is older than the duplication peak of *O. sativa* ($K_s \cong 0.68$). Assuming the ancient WGD event is shared among the floating species, evidence of the ancient WGD event is missing in *W. australiana*. This result is consistent with rapid genome fractionation in *W. australiana* after WGD."

In the fig. 5C, there is a duplication peak of *W. australiana* which is absent in *S. polyrhiza* and younger than the duplication peak of *O. sativa*. Is this contributing to the larger genome size of *W. australiana* than *S. polyrhiza*. In-depth anaysis are suggested.

7. Line 306-309 "A total of 636 differentially expressed genes (DEGs) were identified based on the adjusted p-values. Among these, 218 and 418 genes were upregulated in the floating and submerged phase, respectively (Fig. 6A, Fig. 6B and Table S7)."

The number of DEGs in fig. 6b (3362+74) is inconsistent with the number in the text (636).

8. In the Fig 1. Micrographs of floating and submerged phase of *W. australiana* could have been provided.

Reviewer #2 (Remarks to the Author):

It is known that *Wolffia*, as a watermeal, is rich in protein with fast growth, whereas the absence

of the genome sequence inhibits to understand its genetic infrastructure. Here, the *Wolffia* genome with a relatively small genome size was assembled by reliable long read sequencing with good contiguity and quality. The team also further dissect gene families involved in root and stomata development, anaerobic life under water, and disease resistance, which help us to understand its important biological traits. Still, I have two major concerns as following.

Major revision:

- 1) The same genome for *Wolffia australiana* was published recently (Genome and time-of-day transcriptome of *Wolffia australiana* link morphological minimization with gene loss and less growth control. Todd P. Michael, et.al. *Genome Res.* December 23, 2020, doi:10.1101/gr.266429.120). The novelty need be re-considered. The improvement or the uniqueness of the manuscript compared to the published one should be highlighted. I found that there was a disagreement with the number of protein coding gene (15,000 VS 32,457).
- 2) The analysis of gene family involved in stomata, disease resistance, and adaption are critical biological features of the watermeal, but it is almost pure bioinformatic speculation without further experimental validation. It is not persuasive in terms of deduced conclusions.

Minor revision

- 1) Gene prediction and assessment: please describe the whole picture of *Wolffia* genome annotation including gene number, gene length in Main text.
- 2) Page 8 line 160: A typo of "shed".
- 3) Figure 3B: Please change the gene name into a strict format.
- 4) Page 8 line 171: Gene family expansion of water floating plants. Page 12 line 266: "gene family expansion in *Wolffia australiana*". Although different gene families were mentioned, the results were not organized well. Please arrange the Results more logically and smoothly.
- 5) RNA-seq and other analysis methods are missing.

Reviewer #3 (Remarks to the Author):

This manuscript describes a deep-sequencing genome assembly of a very unique floating plant *Wolffia australiana*. By using gene clustering, pathway mapping, synteny analysis, and other comparative approaches, the authors found extensive functional gene loss and gene expansion during the rapid evolution of the unique species. The authors also did RNAseq analysis to explore the mechanisms how this plant survive winter by comparing the gene regulation during floating and submerged phases. The manuscript is well written despite some minor mistakes below. I cannot find big flaws of this manuscript. The only concern I have for this paper is that the limited economic value and the simple genome of *Wolffia Australiana* would decrease the application and impact of this study.

Line 111: for the first time the Latin name of a spices appears, use the full name *Oryza sativa*, for the rest, the genus name can be initialed *O. sativa*. Same roles for the other spices.

Fig. 2A: The entire Latin name is always italicized; if italics are not possible, the alternative way is to underline both the genus name and the specific epithet.

Reviewer #1 (Remarks to the Author):

1. In the portion of “Root degeneration of *Wolffia australiana*”. The reduction or absence of auxin and gibberellin pathways were proposed to explain the absence of root and shoot growth. In table s4, “ENOG411DVJU” and “ENOG411DV05” involved in the pathway of cytokinin were missing. Additional analysis of reduction/expansion of cytokinin and other hormone pathways will make the explain more convincing.

As this reviewer suggested, we checked if the cytokinin signaling pathway is impaired in the genome of *W. australiana*. The ENOG411DV05 (cytokinin oxidase, CKX7), ENOG411DVJU (CKX4, CKX3, CKX2) gene families are absent in this genome and they have been reported to degrade cytokinin. Genes known to participate in the cytokinin signaling pathway are not highly conserved among Viridiplantae. Nonetheless, we suggest that *W. australiana* is missing regulatory components in the cytokinin pathway. From this observation, the unique morphology of *Wolffia* including lack of root and leaf may be explained by our findings regarding the cytokinin signaling pathway. We added this discussion in the main text as follows (LN. 250):

“Furthermore, the cytokinin signaling pathway in the genome of *W. australiana* appears to be disrupted (Fig. 4A). For instance, cytokinin oxidase (CKX) that degrades cytokinin is missing. As a consequence, increased cytokinin response could inhibit lateral root development³⁹. In addition, large parts of cytokinin signaling and regulatory components, such as response regulators (Type A-RR, Type B-RR) and histidine phosphotransferase proteins (AHP) are absent from the genome⁴⁰. Based on this finding, we postulate that *W. australiana* may be unable to control cytokinin regulation, resulting in root degeneration.”

Furthermore, the other rootless species should be compared, such as *Utricularia gibba*.

We expect the evolution of *U. gibba* to be very different from that of the monocot plant *W. australiana* and sister water floating species. *U. gibba*'s morphological feature of indistinguishable tissues earned it the description of "rootless," whereas *W. australiana* lacks a root. Nonetheless, we ran Egnog with the protein sequences from *U. gibba* (Genome id58573, CoGe) and found that LBD29, WOX5, RSL1 are also missing. We added an explanation at LN. 396 as follows.

“Interestingly, in the genome of the rootless eudicot *Utricularia gibba*⁶¹, auxin signaling pathway components, including LBD29, WOX5 and RSL1, are missing.”

2. Line 30-33 : “Watermeal, *Wolffia australiana*, is the smallest known flowering monocot that is rich in protein. Despite its great potential as a biotech crop, basic research on *Wolffia* is in its infancy and a high-quality genome assembly does not yet exist.”

This is not true anymore. The genomes of two *W. australiana* strains (7733, 8730) have been published recently (doi: 10.1101/gr.266429.120). The sentence have to change a tiny bit.

We removed the “and a high-quality genome assembly does not yet exist.” at LN. 31.

“Despite its great potential as a biotech crop, basic research on *Wolffia* is in its infancy.”

3. Line 182-183: “The alcohol dehydrogenase cluster (ENOG411DZ78) contained *W. australiana* genes that were mostly tandemly-duplicated (Fig. 3B).” The direct evidence for “tandemly-duplicated” should be provided. Then, in Fig. 3b: the genes should be marked with distinct color one by one for those involved in alcohol dehydrogenase cluster of *W. australiana*, *S. polyrhiza*, and *N. colorata*. Some genes were ignored while some were marked improper, such as SPO_Spipo4G0092300, ZmA_KMZ75204.1, SPO_Spipo9G0046300, GWHPAAYW008765...

The cases of tandem duplication can be readily recognized by the gene IDs that indicate the corresponding scaffold and gene order (e.g., Waus00499g15444, Waus00499g15445). As this reviewer requested out, we have changed the color of each aquatic species name in Figure 3B. We changed figure legend in LN. 716 as follows.

“The blue line indicates aquatic plants with their species names being distinguished by different colors. Tandemly duplicated genes in the *W. australiana* genome are indicated by the green box. The gene IDs, which indicate the corresponding scaffold and gene order number, can be used to identify tandemly duplicated loci.”

4. Line 250-252: “Although NLRs are not present in green algae (e.g., *Chlamydomonas reinhardtii*), bryophytes (*Physcomitrella patens*) possess 49 NLRs, and *A. thaliana* and *O. sativa* contain 182 and 438 NLRs, respectively.” Add the information of other aquatic plants, such as *S. polyrhiza*, *Z. marina*, and *N. colorata*.

We added the number of NLRs in *S. polyrhiza* and *Z. marina* in LN. 278.

“In the case of aquatic plants, they have been reported to have 44, 52, and 416 NLRs for *Z. marina*, *S. polyrhiza*, and *N. colorata*, respectively.”

5. Line 273-275: “Together with the estradiol 17-dehydrogenase expansion of water floating plants, cuticle wax biosynthesis would be important for the survival of *W. australiana*.” This is an interesting finding, please provide in-depth analysis of the relationship between the cuticle wax and the survival of *W. australiana*. i.e. protective layers against damaging UV irradiation/pathogen infection, hydrophobicity of granule frond, and keep the correct position.

We added a possible explanations for wax biosynthesis and survival of *Wolffia* in LN. 302.

“i) the capacity of wax components to repel water would be important for *W. australiana* to float in the correct position with the stomata facing the air to aid respiration, ii) wax cuticle layers can minimize UV radiation damage and improve fitness even when exposed to direct sunlight on the water surface with no shade, and iii) it is suggested that plant cuticles can play significant roles in plant-pathogenic and nonpathogenic relationships. Collectively, these expanded gene families are likely to play a critical role in the adaptation to an aquatic environment.”

6. Line 292-297 “We calculated the mean K_s value distribution of intra-genome syntenic blocks in *W. australiana* and in sister species (Fig. 5C). The floating species, *N. colorata* and *S. polyrhiza* often retain the ancient whole genome duplication (WGD) peak ($K_s \cong 0.85$), which is older than the duplication peak of *O. sativa* ($K_s \cong 0.68$). Assuming the ancient WGD event is

shared among the floating species, evidence of the ancient WGD event is missing in *W. australiana*. This result is consistent with rapid genome fractionation in *W. australiana* after WGD.” In the fig. 5C, there is a duplication peak of *W. australiana* which is absent in *S. polyrhiza* and younger than the duplication peak of *O. sativa*. Is this contributing to the larger genome size of *W. australiana* than *S. polyrhiza*. In-depth analysis are suggested.

The total number of synteny blocks with a mean K_s value < 0.1 was 134, with 989 gene pairs. It is difficult to generalize the evolution of genome size from the recent duplication of genes. Rather, we believe that the difference in genome size is due to changes in repeat content. We added an explanation in detail in LN. 331.

“Furthermore, a very recent WGD peak in *W. australiana* was found near $K_s \cong 0.02$. The total number of synteny blocks with a mean K_s value < 0.1 was 134 and they consisted of 989 gene pairs. These recent gene duplications may be explained by selection to functional versatility, or alternatively, they may be more ancient events that are being homogenized by concerted evolution”

7. Line 306-309 “A total of 636 differentially expressed genes (DEGs) were identified based on the adjusted p -values. Among these, 218 and 418 genes were upregulated in the floating and submerged phase, respectively (Fig. 6A, Fig. 6B and Table S7).” The number of DEGs in fig. 6b (3362+74) is inconsistent with the number in the text (636).

Thank you for finding this mistake. We corrected the number of DEGs, increasing it to 698 because we found some DEGs were dropped during data processing. We corrected this number in the main text, Table S7, Fig. 6 accordingly (LN. 344) and corrected the number in Line 344.

“A total of 698 differentially expressed genes (DEGs) were identified based on the Bonferroni adjusted p -values. Among these, 189 and 509 genes were upregulated in the floating and submerged phase, respectively (Fig. 6A, Fig. 6B and Table S7).”

8. In the Fig 1. Micrographs of floating and submerged phase of *W. australiana* could have been provided.

We added a micrograph of *W. australiana* to Fig 1 showing the chloroplast and adaxial stomata (figure legend in LN. 706 as follows).

“C. Micrograph of *W. australiana* showing the presence of stomata on the adaxial side and the absence of stomata on the abaxial side.”

Reviewer #2 (Remarks to the Author):

It is known that *Wolffia*, as a watermeal, is rich in protein with fast growth, whereas the absence of the genome sequence inhibits to understand its genetic infrastructure. Here, the *Wolffia* genome with a relatively small genome size was assembled by reliable long read sequencing with good contiguity and quality. The team also further dissect gene families involved in root

and stomata development, anaerobic life under water, and disease resistance, which help us to understand its important biological traits. Still, I have two major concerns as following.

Major revision:

1) The same genome for *Wolffia australiana* was published recently (Genome and time-of-day transcriptome of *Wolffia australiana* link morphological minimization with gene loss and less growth control. Todd P. Michael et.al. Genome Res. December 23, 2020, doi:10.1101/gr.266429.120). The novelty need to be re-considered. The improvement or the uniqueness of the manuscript compared to the published one should be highlighted. I found that there was a disagreement with the number of protein coding gene (15,000 VS 32,457).

We are aware of the paper's publication earlier this year; however, we assert that the contents of the genome analyses are different. Although we have submitted the assembly of *W. australiana* genome to NCBI on Mar 12, 2020, we added the comparison of the assembly of Michael et. al. (**Table S2**). As the reviewer pointed out, the number of predicted proteins is higher than that of Michael et. al. and we discussed this in the main text. We separate the number of genes and proteins to avoid confusion between the gene and protein number as follows (**LN. 146-154**). Furthermore, for the genome comparison, we aligned the two assemblies to see their consistency and differences. We found our assembly contains additional 212 scaffolds that are not matched with any Wa7733 scaffolds (**LN. 105-116**).

“LN. 105-116: Comparison with the recently published genomes of *W. australiana* 7733 and 8730 strains showed that our assembly of the *W. australiana* 8730 strain has a higher N50 value. Although use of mate-pair and 10X platform analyses aided in longer scaffolding of the PacBio and Illumina data, additional sequence data are needed to increase completeness of the genome. This is exemplified by the recently published chromosomal-level assembly of *Spirodela* that used Oxford Nanopore and Hi-C methods. We also compared the genome assemblies to determine if they align well and if extra scaffolds or contigs exist (Fig. S2B). Even though there are some size differences between homologous scaffolds, both genomes could be aligned well. There were 212 scaffolds in our assembly that were not paired with any Wa7733 scaffolds totaling ~13 Mbp (Fig. S2C). The scaffold size distribution revealed a modal peak of 60 kbp and the scaffolds contained an additional 325 genes that were classified by KEGG as enzyme, transporter, chromosome and associated proteins, among others (Fig. S2D).

LN. 146-154: The total number of predicted genes is 22,293 and the total number of proteins is 32,457, including splice variants. The average length of proteins was 396 aa, with a standard deviation of 354 aa. We assessed the predicted gene and transcript set using BUSCO and found a high coverage of known conserved genes with >94.7% and >76.3% for Viridiplantae and Liliopsida datasets as complete, respectively (Fig. S3). The results of the BUSCO test on the Liliopsida dataset are comparable to the Wa7733 assembly which showed 69% completeness (Table S2). For the functional annotation and phylogenetic classification of the predicted genes, we used the EggNOG database and a total of 26,205 (80.7%) out of 32,457 predicted proteins were successfully annotated.”

2) The analysis of gene family involved in stomata, disease resistance, and adaption are critical biological features of the watermeal, but it is almost pure bioinformatic speculation without further experimental validation. It is not persuasive in terms of deduced conclusions.

We agree about the reviewer's concerns, and bioinformatic data-based speculation is of lesser value than functional data. However, we point to many genome papers that have recently been published in top journals (e.g., Science and Nature) that took a similar approach. A paper by Michael *et. al.* (Genome Res. 2020) was also almost pure bioinformatic speculation. We also wish to have experimental validation of our hypotheses, however, this will take many years to achieve. We think that our genome paper presents a data-driven analysis that presents possible explanations for plant evolution that help guide current and future research in the field. The human genome project was a remarkable achievement because it provided the DNA blueprint for countless subsequent studies of gene and repeat function. Our work is obviously not of this stature, but we expect that it will nonetheless prove of high interest to the community of plant scientists interested in major evolutionary transitions.

Minor revision

1) Gene prediction and assessment: please describe the whole picture of *Woffia* genome annotation including gene number, gene length in Main text.

We added the explanation of *Wolffia* genome annotation in the main text as suggested (LN. 146).

“The total number of predicted genes is 22,293 and the total number of proteins is 32,457, including splicing variants. The average length of proteins was 396 aa, with a standard deviation of 354 aa.”

2) Page 8 line 160: A typo of “shed”.

Corrected.

3) Figure 3B: Please change the gene name into a strict format.

We have removed unnecessary taxonomy numbers in front of the gene names (Figure 3B).

4) Page 8 line 171: Gene family expansion of water floating plants. Page 12 line 266: "gene family expansion in *Wolffia australiana*". Although different gene families were mentioned, the results were not organized well. Please arrange the Results more logically and smoothly.

To prevent misunderstanding, we renamed the subsection as, "Gene family expansion in *Wolffia australiana* related to environmental survival," and added explanation as follows (LN. 293-308)

“Gene family expansion in *Wolffia australiana* related to environmental survival

In addition to the missing orthologs in *W. australiana*, we found 17 gene families that show a significant increase in copy number (Table S4). The two most notable over-represented EggNog clusters are arabinosyltransferase (XEG113) and O-acyltransferase WSD1-like genes that have 27 and 26 gene copies in *W. australiana*, which is >10-fold higher than in the sister species *S.*

polyrhiza. Arabinosyltransferase (XEG113) participates in the arabinosylation of cell wall components, which is important for cell elongation, root development, and cell wall defense ^{49,50}, whereas O-acyltransferase WSD1-like genes participate in wax biosynthesis. Together with the estradiol 17-dehydrogenase gene expansion in water floating plants, cuticle wax biosynthesis would also be important for the survival of *W. australiana*, because, i) the capacity of wax components to repel water would be important for *W. australiana* to float in the correct position with stomata facing the air to aid respiration, ii) wax cuticle layers can minimize UV radiation damage ²⁹ and improve fitness even when exposed to direct sunlight on the water surface with no shade, and iii) it is suggested that plant cuticles can play significant roles in plant-pathogenic and nonpathogenic relationships ⁵¹. Collectively, these expanded genes are likely critical to adapt to the aquatic environment.”

5) RNA-seq and other analysis methods are missing.

We added RNAseq-related analysis methods in the main text as follows (LN. 493-498).

“RNAseq analysis

Low-quality reads were trimmed using trimmomatic software. Using the program Kallisto, the filtered short reads were mapped to the *W. australiana* reference coding sequences. Deseq2 was used to extract candidate gene set as differentially expressed genes (DEG) that differentiate between the floating and submerged phases using the gene expression counts. The final candidate genes were chosen based on Bonferroni modified *p*-values < 0.001.”

Reviewer #3 (Remarks to the Author):

This manuscript describes a deep-sequencing genome assembly of a very unique floating plant *Wolffia australiana*. By using gene clustering, pathway mapping, synteny analysis, and other comparative approaches, the authors found extensive functional gene loss and gene expansion during the rapid evolution of the unique species. The authors also did RNAseq analysis to explore the mechanisms how this plant survive winter by comparing the gene regulation during floating and submerged phases.

The manuscript is well written despite some minor mistakes below. I cannot find big flaws of this manuscript. The only concern I have for this paper is that the limited economic value and the simple genome of *Wolffia australiana* would decrease the application and impact of this study.

Line 111: for the first time the Latin name of a spices appears, use the full name *Oryza sativa*, for the rest, the genus name can be initialed *O. sativa*. Same roles for the other spices.

Fig. 2A: The entire Latin name is always italicized; if italics are not possible, the alternative way is to underline both the genus name and the specific epithet.

We added the full name as the reviewer suggested, “*Oryza sativa* and *Zostera marina*” (LN. 127, LN. 198). We corrected Fig. 2 italicized all species names.

REVIEWERS' COMMENTS:

Reviewer #1 (Remarks to the Author):

The manuscript generated a well-assembled genome of *Wolffia australiana*, explained the adaptation to aquatic environment through the analysis of the gene expansion/missing. The author revised the manuscript responsibly according to the comments. The manuscript now could be accepted for publication upon my opinion.

Reviewer #2 (Remarks to the Author):

In the revised version, the authors compared the genome assembly and annotation with the published one, highlighted the difference and uniqueness. The revised version was highly improved and answered all my concerns. Here are some small issues.

- 1) It was claimed that "Comparison with the recently published genomes of *W. australiana* 7733 and 8730 strains showed that our assembly of the *W. australiana* 8730 strain has a higher N50 value.". Could you please present how much higher of the contig N50 (maybe list the details of *W. australiana* 7733 and 8730).
- 2) Here, "This is exemplified by the recently published chromosomal-level assembly of *Spirodela* that used Oxford Nanopore and Hi-C methods 17,18.". Another good example to support your conclusion is missing in your citation. The contiguity of *Spirodela* genome was significantly improved using Pacbio long read sequencing compared to NGS (Plant evolution and environmental adaptation unveiled by long-read whole-genome sequencing of *Spirodela*, An et.al. PNAS, 2019).
- 3) Line 384: delete "made"
- 4) The spelling of "expanded" is not corrected in Figure 6.